# An Experimental Study of the Surface Roughness of SiCp/Al with Ultrasonic Vibration-Assisted Grinding

**Jie Ying [1], Zhen Yin [1,2,*], Peng Zhang [1], Peixiang Zhou [3], Kun Zhang [1] and Zihao Liu [1]**

1   School of Mechanical Engineering, Suzhou University of Science and Technology, Suzhou 215009, China
2   Suzhou Key Laboratory of Precision and Efficient Machining Technology, Suzhou University of Science and Technology, Suzhou 215009, China
3   School of Intelligent Manufacturing, Tianping College, Suzhou University of Science and Technology, Suzhou 215009, China
*   Correspondence: yinzhen12@mail.usts.edu.cn; Tel.: +86-15599019889

**Abstract:** Due to the differences in mechanical properties of Al and SiC particles, the problems of SiC particle pullout and high surface roughness will occur in the processing of SiCp/Al composites. However, the ultrasonic vibration-assisted grinding of SiCp/Al can effectively decrease the appearance of such problems. A comparative experimental study of the ultrasonic vibration-assisted and ordinary grinding of SiCp/Al is conducted. First, the effect of ultrasonic amplitude on the removal form of SiC is summarized by observing the surface morphology of the sample. Then, the primary reasons for the pullout of SiC particles and high surface roughness in SiCp/Al processing are analyzed. The variation law of the surface roughness of SiCp/Al under different ultrasonic amplitudes and grinding parameters is summarized through a single-factor experiment. The results show that ultrasonic vibration-assisted grinding is beneficial for reducing the surface roughness of SiCp/Al. When grinding linear speed of grinding wheel $v_s$ increases from 2.512 m/s to 7.536 m/s, surface roughness $R_a$ decreases from 0.25 μm to 0.16 μm. when feed rate $v_w$ increases from 100 mm/min to 1700 mm/min, surface roughness $R_a$ increases from 0.13 μm to 0.20 μm. When grinding depth $a_p$ increases from 0.01 mm to 0.05 mm, surface roughness $R_a$ increases from 0.13 μm to 0.19 μm. When ultrasonic amplitude $A$ is increased from 0 μm to 2 μm, surface roughness $R_a$ decreases from 0.26 μm to 0.15 μm. When ultrasonic amplitude $A$ is increased from 2 μm to 4.4 μm, surface roughness $R_a$ increases from 0.15 μm to 0.18 μm.

**Keywords:** SiCp/Al; ultrasonic vibration assisted grinding; surface topography; surface roughness; SiC particle

## 1. Introduction

Silicon carbide (SiC)-reinforced aluminum (Al) matrix (SiCp/Al) is a particle-reinforced metal matrix composite material composed of Al and SiC [1,2]. It combines the advantages of an Al alloy matrix and the characteristics of SiC particles. Consequently, SiCp/Al exhibits excellent properties, such as high specific strength, good plastic workability, low density, high hardness, and low thermal expansion coefficient; it is widely used in industrial manufacturing fields, such as aviation, aerospace, and automobile [3,4].

However, the differences in mechanical properties of Al and SiC particles make machining SiCp/Al materials relatively difficult. Conventional machining has many problems, such as short tool life, large cutting force, and low machining efficiency; pulling out SiC particles and forming pits are easy, resulting in high surface roughness [5,6]. With the expansion of the application range of SiCp/Al and the improvement of the performance requirements for modern high-end equipment, precision machining methods for SiCp/Al have elicited the attention of many scholars. Yinet al. [7] observed a series of surface morphologies of SiCp/Al and found pits and delamination on the machined surface during the grinding process of SiCp/Al, affecting machining quality. They optimized grinding



process parameters and obtained SiCp/Al with a surface roughness of 0.6 μm. Recent studies have shown that ultrasonic vibration-assisted grinding technology has achieved good results in improving the surface quality of composite materials. Feng et al. [8] conducted a comparative test of ultrasonic-assisted and ordinary scratching on SiCp/Al. Their test showed that ultrasonic-assisted scratching can improve the damage caused by SiC particles on the surface of a workpiece; it also exhibits an evident effect on improving surface morphology. Zhu et al. [9] established a finite element model of the grinding process of SiCp/Al; its maximum grinding temperature was consistent with the theoretical maximum grinding temperature. The authors analyzed the grinding temperature field and pointed out that the maximum grinding temperature increased with an increase in wheel speed and grinding depth but decreased with an increase in table speed. The studies of many scholars have proven that the ultrasonic vibration-assisted grinding process exerts a positive effect on improving the surface quality of SiCp/Al. However, research on the surface roughness of precision machined SiCp/Al remains imperfect.

In the current study, ultrasonic vibration-assisted grinding experiments are conducted to solve the problem of high machining surface roughness of SiCp/Al. First, the effect of SiC particles' removal form on surface quality is analyzed during the comparative test between ordinary and ultrasonic vibration-assisted grinding. Then, surface micromorphology after processing is observed. Subsequently, a single-factor experiment is performed to analyze the influences of ultrasonic amplitude and grinding elements on surface roughness. Finally, the processing technology for reducing the surface roughness of SiCp/Al is studied. Through ultrasonic vibration assisted grinding and optimization of processing parameters, the surface roughness of SiCp/Al sample less than $R_a$ 0.2 μm can be obtained on the basis of ensuring processing efficiency.

## 2. Experimental Details

### 2.1. Sample Material

The test material is SiCp/Al. its volume fraction is about 55%, and its size is 25 mm × 10 mm × 5 mm. The physical performance parameters of the material are shown provided in Table 1.

**Table 1.** Physical properties of SiCp/Al.

| Material | Density/(g·cm$^{-3}$) | CTE/($\times 10^{-6}$/K) | Thermal Conductivity/[W·(m·K)$^{-1}$] | Bending Strength/MPa | Tensile Strength/MPa | Elastic Modulus/GPa |
|---|---|---|---|---|---|---|
| SiCp/Al | 3.04 | 7.3 | 175 | 400 | 206 | 258 |

### 2.2. Experimental Setup and Detection Method

Grinding experiments were conducted on a JDVT600-A13S machining center. The maximum spindle speed of the computer numerical control machining center is 24,000 r/min, and the repetitive positioning accuracy is ±0.002 mm. The ultrasonic vibration tool holder is independently developed by our research group. The high-frequency AC signal from the ultrasonic generator is transmitted to the ultrasonic vibration transducer through the radio energy transmission device. The high-frequency AC signal is converted into the vibration displacement of the ultrasonic vibration transducer through the inverse piezoelectric effect of piezoelectric ceramics. The vibration displacement can be amplified by the horn, and ultrasonic amplitude can take effect on the front-end tool. Its ultrasonic vibration frequency is 22 kHz, and its maximum ultrasonic amplitude is 7 μm. The machining tool is a slotted electroplated diamond grinding wheel that helps improve chip removal during grinding. Its base material is 45 steel. It has a diameter of 6 mm, an average abrasive grain size of 90 μm, and a slotted width of 0.5 mm. The machining tool is installed at the front end of the ultrasonic vibration tool holder. The vice is fixed on the workbench of the machining center, and the SiCp/Al sample is clamped in the jaws of the vice. The test platform is illustrated in Figure 1.

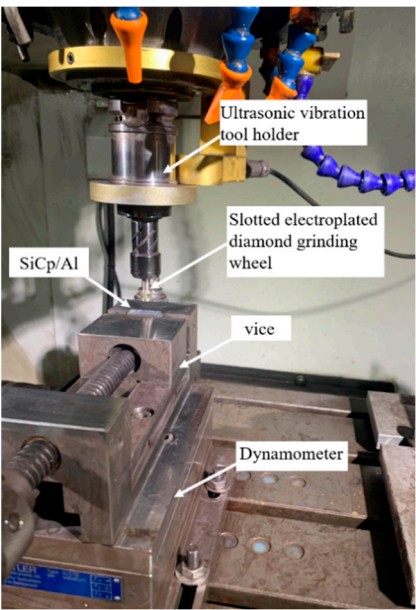

**Figure 1.** Ultrasonic vibration-assisted grinding test platform.

Before measurement, the SiCp/Al sample is cleaned using an ultrasonic cleaning machine to remove particles and stains attached onto the surface of the sample to ensure measurement accuracy. The major damage form of SiC particles on the surface of the machined sample is analyzed with a scanning electron microscope system 2000 times, and the microscopic topography of the machined sample surface is observed 1000 times. A ContourGT-K0 white light interferometer (Bruker, Billerica, MA, USA) is used to observe the 3D morphology of the processed SiCp/Al surface under a 2.5 times mirror and record the number and area of pits on the surface (pits with an area less than 500 μm$^2$ are disregarded during recording). Then, surface roughness is measured. Each machined surface is measured five times, and the average of the five measured values is set as the final surface roughness.

### 2.3. Experimental Design

First, a comparative experiment of ordinary and ultrasonic vibration-assisted grinding is performed. The experimental parameters are as follows: linear speed of grinding wheel $v_s$ = 6.28 m/s, feed rate $v_w$ = 500 mm/min, grinding depth $a_p$ = 0.02 mm, and ultrasonic vibration amplitude $A$ = 2 μm. Then, a single-factor experiment of ultrasonic amplitude and various grinding factors is conducted to analyze the influences of ultrasonic amplitude and various grinding factors on surface roughness $R_a$. The process parameters of grinding are listed in Table 2 [10].

**Table 2.** The process parameters of grinding.

| Experimental Conditions | Parameters |
| --- | --- |
| Linear speed of grinding wheel $v_s$/(m/s) | 2.512, 3.768, 5.024, 6.28, 7.536 |
| Frequency $f$/(kHz) | 22 |
| Ultrasonic vibration amplitude $A$/(μm) | 0, 0.5, 1, 2, 4 |
| Grinding depth $a_p$/(μm) | 0.2, 0.5, 1, 1.5, 2 |
| Grinding width $a_w$/(μm) | 20 |
| Average grain size $q_s$/(μm) | 90 |
| Wheel diameter $d_s$/(mm) | 6 |
| Feed rate $v_w$/(mm/min) | 100, 500, 900, 1300, 1700 |
| Cooling condition | Water-soluble cutting oil with flowrate of 2 L/min and pressure of 2 bar |

## 3. Results and Discussion

### 3.1. Surface Morphologies

For SiCp/Al, the removal method of the SiC particles plays an important role in the surface quality. The major removal methods for SiC particles are extraction, fragmentation, and cutting [11] (Figure 2). When SiC particles are removed through extraction and fragmentation, pits are formed on the surface, resulting in poor surface quality, as shown in Figure 2a,b. When SiC particles are removed through cutting, SiC particles are flat with the surface of the Al matrix, resulting in low surface roughness, as shown in Figure 2c.

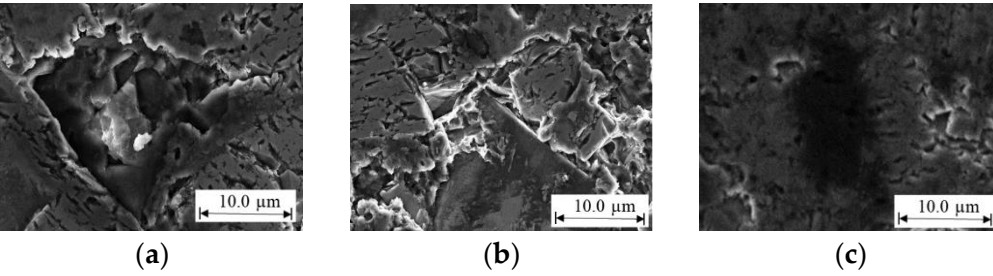

| (a) | (b) | (c) |

**Figure 2.** Major removal methods and morphologies of SiC particles. (**a**) Extraction of SiC particles, (**b**) fragmentation of SiC particles, and (**c**) cutting of SiC particles.

Figure 3 shows the surface micromorphologies of ordinary and ultrasonic vibration-assisted grinding. During ordinary grinding, the Al matrix on the surface of the sample protrudes, and SiC particles are pulled out, resulting in a large number of pits. Consequently, this method increases the surface roughness of the sample, as illustrated in Figure 3a. However, during ultrasonic vibration-assisted grinding, the tool continuously hammers the surface of the sample. Then, the tool performs high-frequency micro-cutting on SiC particles attached onto the surface of the sample. Thus, the SiC particles are removed by cutting [12]. Simultaneously, contact between the tool and the sample surface is intermittent most of the time. This condition is convenient for chip discharge and considerably reduces the cutting force. Consequently, it reduces the phenomenon of SiC particles being pulled out. During ultrasonic vibration-assisted grinding, the removal methods of the Al alloy matrix and SiC particles change, the Al matrix undergoes plastic deformation, the surface becomes flat, and the number of pits is reduced. Therefore, surface roughness is lower than that of ordinary grinding, as shown in Figure 3b.

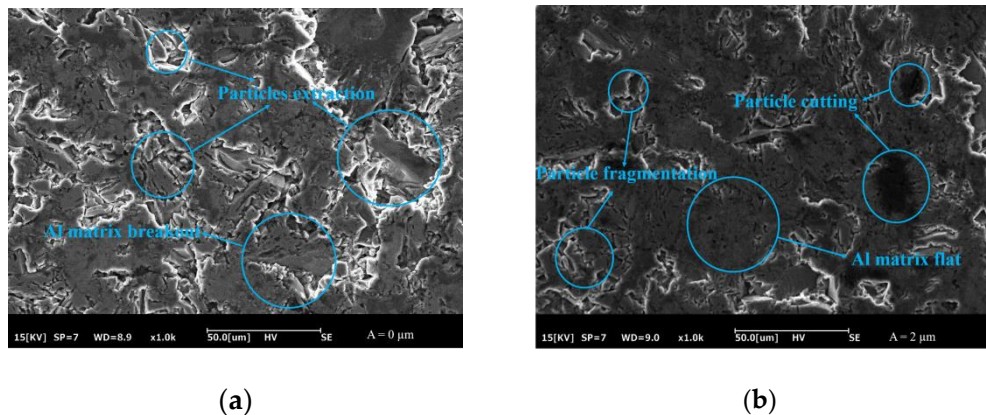

| (a) | (b) |

**Figure 3.** Surface micromorphologies of ordinary and ultrasonic vibration-assisted grinding. (**a**) Ordinary grinding and (**b**) ultrasonic vibration-assisted grinding.

Figure 4 displays the 3D topography of SiCp/Al for ordinary and ultrasonic vibration-assisted grinding. Many large-area pits are found on its surface, indicating that a large number of SiC particles have been pulled out on its surface. Ultrasonic vibration-assisted

grinding of the surface not only reduces the number of pits but also makes the pit area generally smaller. This finding demonstrates that ultrasonic vibration-assisted grinding can effectively perform high-frequency vibration micro-cutting of SiC particles and reduce the number of SiC particles that are pulled out [13]. Figure 5 illustrates the relationship between number of pits and ultrasonic amplitude. Within the range of 6.3 mm × 4.75 mm, 15 pits are formed on the surface during ordinary grinding, and the average pit area is 5700 μm². During ultrasonic vibration-assisted grinding, the number of pits is reduced to 4 when ultrasonic amplitude is 1 μm. This result is 73% less than that of ordinary grinding. The average pit area is reduced to 3000 μm², which is 47% smaller than that of ordinary grinding. When ultrasonic amplitude is 2–4 μm, the number of pits tends to be stable at 2–3, which is 87% less than that of ordinary grinding. The average pit area is 2400 μm², which is 58% smaller than that of ordinary grinding. The preceding results show that ultrasonic vibration-assisted grinding can effectively inhibit the generation of pits and reduce the area of pits. When ultrasonic amplitude is within the range of 2–4 μm, ultrasonic vibration-assisted grinding exhibits a more evident effect on inhibiting the generation of pits.

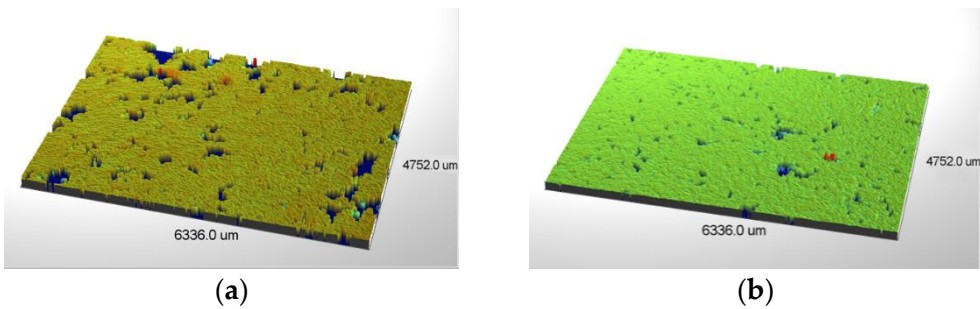

**Figure 4.** 3D topography of SiCp/Al: (**a**) ordinary grinding and (**b**) ultrasonic vibration-assisted grinding.

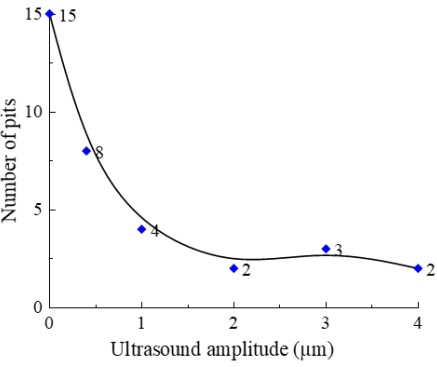

**Figure 5.** Relationship between ultrasonic amplitude and number of pits.

*3.2. Surface Roughness*

Figure 6 shows the relationships between surface roughness and each grinding factor after ultrasonic vibration-assisted grinding.

As shown in Figure 6a, when grinding linear speed of grinding wheel $v_s$ increases from 2.512 m/s to 7.536 m/s, surface roughness $R_a$ decreases from 0.25 μm to 0.16 μm, because the shearing effect of the diamond abrasive particles on the tool for SiC particles increases with an increase in grinding linear speed of grinding wheel $v_s$ [14]. The force in the feed direction is reduced so as not to exceed the bonding force between SiC particles and Al matrix. More SiC particles on the sample surface are removed by cutting. The number of pits on SiCp/Al surface is reduced and the surface is smooth, reducing surface roughness [15].

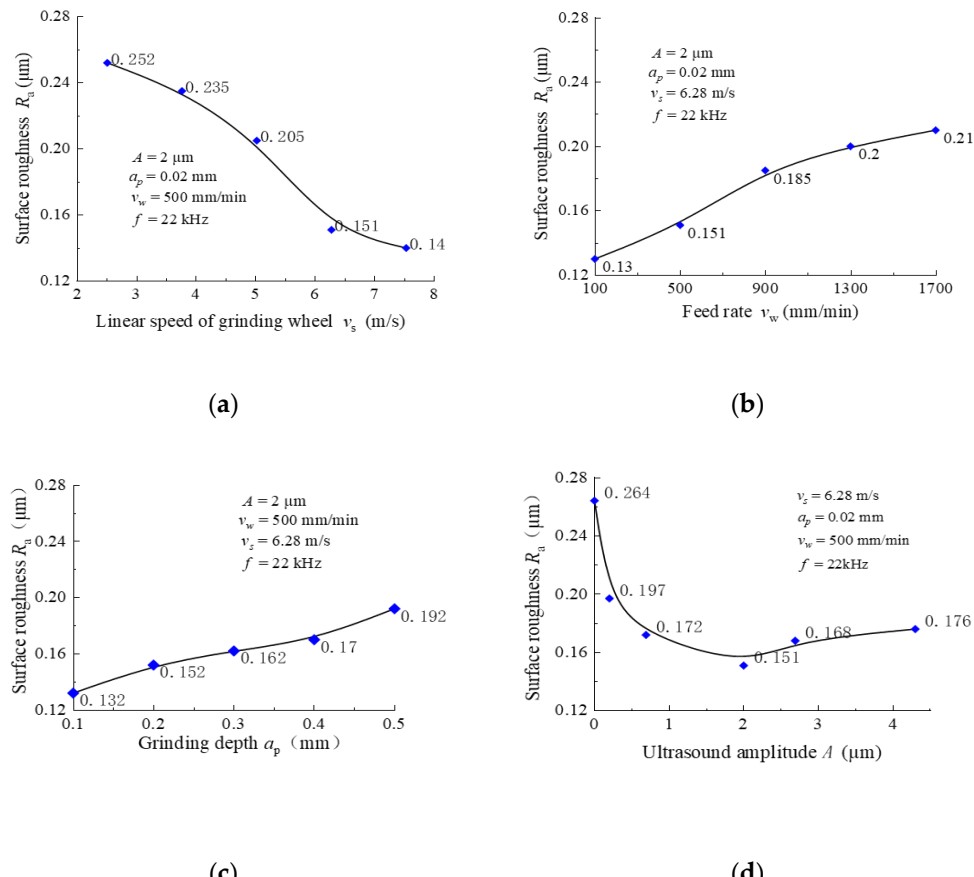

**Figure 6.** Relationships between surface roughness and each grinding factor after ultrasonic vibration-assisted grinding. (**a**) linear speed of grinding wheel and surface roughness, (**b**) feed speed and surface roughness, (**c**) grinding depth and surface roughness, and (**d**) ultrasonic amplitude and surface roughness.

As illustrated in Figure 6b, when feed rate $v_w$ increases from 100 mm/min to 1700 mm/min, surface roughness $R_a$ increases from 0.13 µm to 0.20 µm, because a larger feed rate $v_w$ will lead to higher residual stress, reducing the bonding effect between SiC particles and the Al alloy matrix. Consequently, SiC particles are easier to pull out. More pits appear on the surface of SiCp/Al, resulting in an uneven surface. Thus, surface roughness is increased [16].

As depicted in Figure 6c, when grinding depth $a_p$ increases from 0.01 mm to 0.05 mm, surface roughness $R_a$ increases from 0.13 µm to 0.19 µm, because contact time between abrasive particles and the sample is prolonged as grinding depth $a_p$ increases. Moreover, the number of abrasive particles that participate in grinding in the grinding area increases, increasing the overall grinding force. This will cause the force in the feed direction to exceed the bonding force between SiC particles and Al matrix. This condition will make SiC particles more easy to pull out. Pits are formed on the surface, resulting in uneven surface. Thus, surface roughness increases [17].

As shown in Figure 6d, when ultrasonic amplitude $A$ is increased from 0 µm to 2 µm, surface roughness $R_a$ decreases from 0.26 µm to 0.15 µm. When ultrasonic amplitude $A$ is increased from 2 µm to 4.4 µm, surface roughness $R_a$ increases from 0.15 µm to 0.18 µm. When ultrasonic amplitude $A$ is 2 µm, the minimum surface roughness $R_a$ is 0.15 µm. These results show that the surface roughness $R_a$ of the sample can be deceased with an increase in ultrasonic amplitude $A$. However, when ultrasonic amplitude $A$ exceeds 2 µm, the cracks generated by SiC particles expand, resulting in the fragmentation of SiC particles [18]. The surface of SiCp/Al will produce protrusions, resulting in uneven surface. Thus, surface roughness increases [19].

In summary, surface roughness $R_a$ tends to decrease with an increase in grinding linear speed of grinding wheel $v_s$. As feed speed $v_w$ and grinding depth $a_p$ increase, surface roughness $R_a$ tends to increase. As ultrasonic amplitude $A$ increases, surface roughness $R_a$ initially decreases and then increases. Among the grinding factors, the variation in grinding depth $a_p$ exerts the least effect on surface roughness $R_a$, whereas the variation in ultrasonic amplitude $A$ exhibits the greatest effect on surface roughness $R_a$ [20]. From the perspective of synthesizing surface roughness and machining efficiency, in the ultrasonic vibration-assisted grinding of SiCp/Al, the preferred grinding parameters are ultrasound amplitude $A$ = 2 μm, linear speed of grinding wheel $v_s$ = 6.28 m/s, feed rate $v_w$ = 500 mm/min, and grinding depth $a_p$ = 0.02 mm. The surface roughness $R_a$ = 0.151 μm can be obtained under these preferred grinding parameters.

## 4. Conclusions

In this study, through the comparison of ordinary and ultrasonic vibration-assisted grinding, the damage of SiC particles under the two processing methods is analyzed and the influences of grinding factors on the surface roughness of ultrasonic vibration-assisted grinding of SiCp/Al is discussed. Compared with ordinary grinding, ultrasonic vibration-assisted grinding SiCp/Al can obtain the surface roughness of $R_a$ 0.15 μm on the basis of ensuring the machining efficiency. The major conclusions derived from this study are as follows.

(1) The comparison between ordinary and ultrasonic vibration-assisted grinding shows that the Al matrix in the material undergoes plastic deformation under the action of ultrasonic vibration, and its surface becomes flat and smooth. SiC is hammered to create microcracks, and SiC particles are more easily removed by severing. The use of ultrasonic vibration-assisted grinding for SiCp/Al composite materials is beneficial for reducing the pullout of SiC particles and decreasing the surface roughness of the sample.

(2) Within the range of 6.5 mm × 4.5 mm, the number of pits is 15 during ordinary grinding, and the average pit area is 5700 μm². In ultrasonic vibration grinding, when ultrasonic amplitude is 1 μm, the number and area of pits are significantly reduced. When ultrasonic amplitude is 2–4 μm, the number of pits tends to be stable at 2–3, and the average pit area is 2400 μm², which are 87% and 58% lower than those of ordinary grinding, respectively. These results show that ultrasonic vibration-assisted grinding can significantly reduce the number and area of pits.

(3) Ultrasonic amplitude $A$ exerts the greatest influence on the surface roughness $R_a$ of the SiCp/Al sample, followed by feed speed $v_w$, and grinding linear speed of grinding wheel $v_s$. Conversely, grinding depth $a_p$ exhibits the smallest influence. Surface roughness decreases with an increase in linear speed of grinding wheel $v_s$ and increases with an increase in feed rate $v_w$ and grinding depth $a_p$. Surface roughness $R_a$ initially decreases and then increases with an increase in ultrasonic amplitude. Surface roughness $R_a$ can reach a minimum value of 0.151 μm when ultrasonic amplitude $A$ is 2 μm.

(4) The comprehensive consideration of the surface roughness and machining efficiency of the SiCp/Al sample indicates that the preferred grinding parameters in the ultrasonic vibration-assisted grinding of SiCp/Al are as follows: ultrasound amplitude $A$ = 2 μm, linear speed of grinding wheel $v_s$ = 6.28 m/s, feed rate $v_w$ = 500 mm/min, and grinding depth $a_p$ = 0.02 mm. The surface roughness $R_a$ of SiCp/Al = 0.151 μm can be obtained under these preferred grinding parameters.

**Author Contributions:** Conceptualization, Z.Y.; methodology, Z.L.; validation, K.Z.; investigation, J.Y.; resources, P.Z. (Peixiang Zhou); writing—original draft preparation J.Y.; writing—review and editing, J.Y., Z.Y. and P.Z. (Peng Zhang); project administration, Z.Y. All authors have read and agreed to the published version of the manuscript.

**Funding:** This research was funded by the National Natural Science Foundation of China (grant No. 51905363), the Natural Science Foundation of Jiangsu Province (grant Nos. BK20190940 and BK20210866) and the Natural Science Foundation of the Jiangsu Higher Education Institutions of China (grant Nos. 19KJB460008 and 21KJB460021), the China Postdoctoral Science Foundation (grant No. 2019M661914).

**Institutional Review Board Statement:** Not applicable.

**Informed Consent Statement:** Not applicable.

**Data Availability Statement:** The data presented in this study are available within the article.

**Conflicts of Interest:** The authors declare no conflict of interest.

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
