# Peer review of "An Experimental Study of the Surface Roughness of SiCp/Al with Ultrasonic Vibration-Assisted Grinding"

_metals, doi:10.3390/met12101730_

Round 1

Reviewer 1 Report

The reviewer comments of the paper «Experimental Study on the Surface Roughness of SiCp/Al with Ultrasonic Vibration-assisted Grinding» - Reviewer

The authors presented an article «Experimental Study on the Surface Roughness of SiCp/Al with Ultrasonic Vibration-assisted Grinding». However, there are several points in the article that require further explanation.

Comment 1:

The abstract needs to be improved.

Demonstrate in the abstract novelty, practical significance. Add quantitative and qualitative work results to the abstract. Describe the input and output parameters investigated in the work. It is necessary to describe the most results of the article.

Comment 2:

The introduction needs to be improved.

Now the list of references needs to be supplemented with at least 3-4 more references published over the last 2020-2022 years.

After analyzing the literature, show before formulating the goal of the "blank" spots. Which has not been previously done by other researchers. You must show the importance of the research being undertaken. Show what will be the new research approach in this article. You need to show a hypothesis.

Demonstrate in the abstract novelty, practical significance.

Add a clear purpose to the article.

Comment 3:

2. Experimental details

Are all figures original? If not needed appropriate citations and permissions. Refine this for figures throughout the article.

What is the hardness of the workpiece and how was it measured?

Describe the measurement procedure in more detail. At what point in time? How is the measuring setup set up? How many repetitions of measurements? What statistical methods are used to process experimental results? Describe the experimental stand in more detail. What method of experiment planning is used and why?

Replace "Wheel speed vs/(r/min)" with "Rotational speed n/(rpm)".

Comment 4:

3. Results and discussion

Check the horizontal axis of figure 6a. Size does not match. Check for all figures.

The description of all figures in the text must be supplemented. Minimum 4-5 sentences. It is also important to add a figure with output curves from cutting data. Analyze the nature of these curves in accordance with the influence of the cutting mode on these curves, feed, cutting speed, depth of cut. This needs to be explained in terms of cutting physics. What is the difference from previous work in this area?

Comment 5:

It will be useful to add a section of Nomenclature in which to sign all the physical quantities and abbreviations encountered in the article. There are many physical quantities in the text and such a section will help to find the description of the necessary element.

For example,

p :              Density (g/cm3)

SiC :          Silicon carbide

etc.

Comment 6:

Conclusions needs to be improved.

It is necessary to more clearly show the novelty of the article and the advantages of the proposed method. What is the difference from previous work in this area? Show practical relevance.

The article is interesting, but needs to be improved. Authors should carefully study the comments and make improvements to the article step by step. After major changes can an article be considered for publication in the "Metals".

Reviewer 2 Report

Generally, a nice study which covers the improvement in performance when ultrasonically grinding materials.  

It would be better to use a different word than dandruff. Dandruff is more related to the human body. Perhaps particles, etc would be better.

Abstract could do with quantification to add depth.

It would be better to give cutting speed than rotational speed.

It is vice/vise rather than flat tongs.

How was the ultrasonic amplitude made? A comment regarding this would be good. 

It would be better if the manuscript was written in the third person.

Figures should have the ultrasonic amplitude next to them.

Figure 4 could be improved with better scale bars, etc. The units are missing on the right hand axis as well.

Additional studies could be referenced in the results and discussion to explain the findings. The conclusions could be updated to provide this as well.

Further wider range of references would be preferable.

What was the flowrate and pressure of the grinding fluid.

What was the cut-off length and evaluation length for the surface roughness?

Why was 22kHz selected for the ultrasonic frequency?

Reviewer 3 Report

Suggestions for improving the manuscript are as follows:

1. Please include specific and quantitative results in your Abstract.

2. Keywords should be corrected. "Experimental study" is a cliché.

3. References are representative, but few. Expand your current status overview with new references to highlight your contribution.

4. How you chose the test material and how you chose its dimensions.

5. The density is repeated in the text and in Table 1.

6. Show a figure of the ultrasonic vibration tool holder and briefly explain the principle of operation.

7. How parameters were selected during grinding (Table 2). Why are these parameters representative?

8. Experimental results could be presented in a table. It will be easier to follow.

9. What experimental plan was used? Explain the choice.

10. Can the results obtained be statistically processed? (ANOVA, RSM, etc.)

11. What is the influence of initial surface roughness on the obtained results?

12. Compare your results with previous similar research results.

13. Does your methodology have any shortcomings and/or limitations?

14. What are the directions of future research?

Round 2

Reviewer 1 Report

The authors have improved the article in accordance with the comments. Now the article can be published.

Reviewer 3 Report

The manuscript has been corrected